# Echocardiographic Strain Abnormalities Precede Left Ventricular Hypertrophy Development in Hypertrophic Cardiomyopathy Mutation Carriers

**DOI:** 10.3390/ijms25158128

**Published:** 2024-07-25

**Authors:** Grazia Canciello, Raffaella Lombardi, Felice Borrelli, Leopoldo Ordine, Suet-Nee Chen, Ciro Santoro, Giulia Frisso, Salvatore di Napoli, Roberto Polizzi, Stefano Cristiano, Giovanni Esposito, Maria-Angela Losi

**Affiliations:** 1Department of Advanced Biomedical Sciences, Federico II University of Naples, Via S Pansini 5, 80131 Napoli, Italy; grazia.canciello@unina.it (G.C.); felice.borrelli@unina.it (F.B.); leopoldo.ordine@unina.it (L.O.); ciro.santoro@unina.it (C.S.); salvatore.dinapoli@unina.it (S.d.N.); roberto.polizzi@unina.it (R.P.); stefano.cristiano@unina.it (S.C.); espogiov@unina.it (G.E.); losi@unina.it (M.-A.L.); 2Department of Medicine, University of Colorado Anschutz Medical Campus, Aurora, CO 80045, USA; suet.chen@cuanschutz.edu; 3Department of Molecular Medicine and Medical Biotechnology, Federico II University of Naples, 80131 Naples, Italy; giulia.frisso@unina.it

**Keywords:** hypertrophic cardiomyopathy, screening, genetics, subclinical detection, strain echocardiography, global longitudinal strain, diastolic strain rate, left ventricular hypertrophy

## Abstract

Hypertrophic cardiomyopathy (HCM) is a genetic disease characterized by unexplained left ventricular hypertrophy (LVH), diastolic dysfunction, and increased sudden-death risk. Early detection of the phenotypic expression of the disease in genetic carriers without LVH (Gen+/Phen−) is crucial for emerging therapies. This clinical study aims to identify echocardiographic predictors of phenotypic development in Gen+/Phen−. Sixteen Gen+/Phen− (one subject with troponin T, six with myosin heavy chain-7, and nine with myosin-binding protein C3 mutations), represented the study population. At first and last visit we performed comprehensive 2D speckle-tracking strain echocardiography. During a follow-up of 8 ± 5 years, five carriers developed LVH (LVH+). At baseline, these patients were older than those who did not develop LVH (LVH−) (30 ± 8 vs. 15 ± 8 years, *p* = 0.005). LVH+ had reduced peak global strain rate during the isovolumic relaxation period (SRIVR) (0.28 ± 0.05 vs. 0.40 ± 0.11 1/s, *p* = 0.048) and lower global longitudinal strain (GLS) (−19.8 ± 0.4 vs. −22.3 ± 1.1%; *p* < 0.0001) than LVH- at baseline. SRIVR and GLS were not correlated with age (overall, *p* > 0.08). This is the first HCM study investigating subjects before they manifest clinically significant or relevant disease burden or symptomatology, comparing at baseline HCM Gen+/Phen− subjects who will develop LVH with those who will not. Furthermore, we identified highly sensitive, easily obtainable, age- and load-independent echocardiographic predictors of phenotype development in HCM gene carriers who may undergo early preventive treatment.

## 1. Introduction

Hypertrophic cardiomyopathy (HCM) is a hereditary disease mainly caused by mutations in genes encoding sarcomeric proteins, exhibiting unexplained left ventricular (LV) hypertrophy (in absence of abnormal loading conditions), normal or supernormal ejection fraction, and diastolic dysfunction [1]. Accordingly, with the guidelines [1] being the LV hypertrophy (LVH) as typically asymmetrical in HCM, the presence of LVH is defined by the presence of a maximal wall thickness (MWT) ≥ 13 mm in the familial forms. The histopathology of HCM is characterized by cardiomyocyte hypertrophy and disarray, interstitial fibrosis and intramyocardial small-vessel disease [2,3,4]. Myocyte disarray occupying more than 10% of the myocardium is broadly accepted as a diagnostic hallmark for the histological diagnosis of HCM [5,6]. Moreover, disarray is considered the earliest response to the structural and/or functional abnormalities induced at cellular and subcellular level by the mutated sarcomeric protein [7]. Disarray has also been shown to be a marker of increased risk of sudden cardiac death [8].

Disease-causing mutations are detected in about half of HCM probands, with an identifiable pathogenic variant in myosin heavy chain 7 (*MYH7*) and myosin-binding protein C 3 (*MYBPC3*) accounting for about 75% of the genotype positive cases [9]. Advances in genetic testing have improved the knowledge of the molecular causes of HCM, facilitating the development of novel treatments. Furthermore, the significant reduction in the costs has allowed the introduction of cascade genetic family screening in the clinical practice. The genetic screening of the family members of the proband may identify mutation-carriers without overt HCM, named “genotype-positive/phenotype-negative” (Gen+/Phen−) subjects [10]. Gen+/Phen− individuals offer the opportunity to study cardiac abnormalities before the development of hypertrophy, providing insights into the earliest biomechanical defects caused by the presence of the causal mutation [10]. Cross-sectional studies have shown that Gen+/Phen− subjects have hyperdynamic LV contraction and diastolic dysfunction when compared to controls [11,12,13]. In these studies, systolic function has been assessed by measuring the LV ejection fraction by 2D, whereas diastolic dysfunction has been detected by measuring the Tissue Doppler (TD) velocities by Doppler echocardiography; in particular, it has been shown that Gen+/Phen− individuals have lower TD velocities, indicative of increased diastolic pressures, as compared with normal subjects [12]. Only in one study were Gen+/Phen− subjects evaluated at baseline and after two years of follow-up; this study showed that among 12 Gen+/Phen−, whose parameters of diastolic dysfunction were lower than controls, 6 developed the phenotype, suggesting that a subtle diastolic dysfunction could predict development of the phenotype [12]. However, differences between Gen+/Phen− who develop LV hypertrophy (LVH) vs. those who do not develop LVH during the follow-up have never been tested. The identification of early signs of phenotype development could help to identify the moment to switch to a closer follow-up or possibly to start a preventive treatment in the HCM gene carriers. Therefore, we performed a study to assess whether subtle cardiac abnormalities of diastolic and systolic function may predict the development of LVH in HCM Gen+/Phen− subjects.

## 2. Results

### Genetic Screening, Demographics, and Echocardiographic Characteristics

Sixteen family members of twelve HCM probands met the inclusion criteria to be enrolled in the study. We identified 11 pathogenic/likely pathogenic variants in three sarcomeric genes. Six subjects carried mutations in *MYH7*; nine subjects were carriers of mutations in *MYBPC3* (38% and 56% of the study population, respectively) ; only one subject carried a pathogenic variant in the Troponin T (*TNNT2*) gene (6% of the study population). All the mutations identified in *MYH7* and in *TNNT2* were missense mutations, while in *MYBPC3* we identified two missense mutations in three individuals, two of whom were from the same family (33% of the MYBPC3 carriers) and radical mutations in six subjects (67% of the MYBPC3 carriers; two alternative splicing variants in two unrelated individual, one indel insertion–deletion in two related subjects and one nonsense variant in two related subjects). The causal gene, the specific mutation, and the relationship with the probands are shown in Table 1.

During a follow-up of 8 ± 5 years, 5 subjects out of 16 gene carriers developed LVH. 

The distribution of the mutations in the *MYH7*, *MYBPC3* and *TNNT2* and the domains of the proteins which are affected are shown in Figure 1.

*MYH7* encodes for β-Mosin Heavy Chain (β-MHC) and is composed of a N-terminal globular head domain and a C-terminal coiled-coil domain [14]. Most of the mutations we identified in *MYH7* are located in the myosin motor domain; hence, they may potentially affect the binding kinetics of the myosin heads to the thin filaments of actin. One of the mutations we identified, the *p*.R723C, is located in the converter domain of β-MHC and could potentially alter the elastic property of the converter domain, resulting in a stronger power stroke during the crossbridge cycle. However, the functional impacts of these mutations on the pathogenesis of HCM remain to be unraveled. *MYPBPC3* encodes for the protein cMyBPC which regulates the cross-bridge cycle through its biophysical interactions with both β-MHC and actin [15,16]. The mutations in *MYPBPC3* identified in this study are listed as pathogenic by the ClinVar database (https://clinvarminer.genetics.utah.edu/variantsbygene/MYBPC3/condition/not%20provided/pathogenic, accessed on 11 June 2024). The mutations were located in the Immunoglobulin-like (Ig) C2 and C10 domains and in the Fibronectin type-III (Fn.3) domains C6 and C9. The *TNNT2* mutation we identified in one of the carriers is localized in exon 8, near the N terminus of Troponin T. Mutations in *TNNT2* account for approximately 15% of familial HCM and the majority of the missense mutations are located in exons 8–16 [17,18]. Mutations of *TNNT2* are associated with high risk of sudden cardiac death despite mild left ventricular hypertrophy. The majority of *TNNT2* mutations of familial HCM affect Ca^2+^ sensitivity and ATPase activity, resulting in impaired contractile properties of the cardiac myocytes [19].

The localization of the mutations did not show clustering of the carriers who developed vs. those who did not develop the phenotype in our study population.

Baseline differences between subjects who developed LVH vs. those who did not are reported in Table 2.

Patients who developed LVH were older and showed higher MWT at baseline than subjects who did not develop LVH (overall, *p* < 0.03). In addition, carriers who developed LVH showed lower septal E’, higher E/E’ and reduced SRIVR, as compared with carriers who did not develop LVH, indicating that subjects who became Phen+, had diastolic dysfunction already at baseline (overall, *p* < 0.05) (Table 2). Moreover, GLS was significantly lower in patient who developed the phenotype (*p* < 0.0001) (Table 2). Because age should be considered when evaluating the diastolic function [20], we performed a Pearson correlation between age and echocardiographic indexes of diastolic function in our population and found that the E/E’ ratio was directly related to age (*p* = 0.011), whereas SRIVR and GLS were not correlated with age (*p* > 0.08).

We tested variations of SVIVR and GLS during the follow-up. Because the repeatability coefficient was 0.03 1/s for SRIVR and −0.56% for GLS, only variations superior to this cut-off were considered indicative of effective changes. Figure 2 shows the variation from baseline to follow-up of SRIVR and of GLS between patients who developed and those who did not develop the phenotype. However, considering the cut-off of changes, GLS worsened similarly between Phen− and Phen+ at follow-up (*p* = 0.611) (Table 3), whereas SRIVR worsened in overall Phen+ patients, although this prevalence did not reach a significant level (*p* = 0.069) (Table 3).

Of note, overall, Phen+ had a GLS ≥ −20%, whereas, overall, subjects Phen− had GLS < −21%. Finally, the LV ejection fraction was slightly higher at baseline in Phen+; however, at the end of the follow-up it was similar between subjects who developed and those who did not develop the phenotype (Figure 2).

Figure 3 reports examples of a subject who did not develop (left panel) and one subject who developed (right panel) the phenotype during the follow-up.

## 3. Discussion

We were able to retrospectively enroll sixteen Gen+/Phen− subjects harboring a causal mutation for HCM and gain valuable insights into the early myocardial changes assessed by echocardiography preceding the phenotypic development of overt disease. During a follow-up of about 8 years, five subjects progressed to manifesting the HCM phenotype, characterized by increased LV wall thickness above the cut-offs for the diagnosis of familial HCM.

Our advanced echocardiographic assessments, including GLS and SRIVR, provided critical predictive markers of phenotypic development. We observed that subjects with lower diastolic SRIVR and lower systolic deformation, as indicated by GLS at baseline were more likely to develop the HCM phenotype. These suggests that SRIVR and GLS are able to detect subtle myocardial dysfunction, occurring even before the onset of LVH. The reduced SRIVR at baseline suggests the presence of global diastolic myocardial dysfunction [21], indicating impaired myocardial relaxation, a typical feature of HCM pathophysiology. Although in our study other diastolic function indices measured by pulsed Doppler and TDI were also lower in Phen+ subjects compared to Phen− subjects, they may not have a reliable early predictive value in this context. This limitation stems from their close association with age-related changes and their reduced sensitivity in patients with normal cardiac function, as referenced in previous studies [22].

Previous studies aimed to understand echocardiographic abnormalities in Gen+/Phen− subjects. Ho CY et al. performed a cross-sectional study, comparing TDI and strain-echocardiographic data among normal subjects, Gen+/Phen− individuals, and overt HCM patients^13^.The differences the authors identified among the three groups were of great interest, yet did not clarify the characteristics that could help clinicians to predict phenotype development in Gen+/Phen− subjects. Another similar cross-sectional study which focused on MWT and TDI has compared normal subjects, Gen+/Phen− individuals, and overt-HCM patients [23]. The authors found that greater LVMWT was associated with more prominent cardiac abnormalities in a continuous, although not always linear, manner, and that a single value of LVMWT could not dichotomize the presence or absence of the disease. Another study characterized the early manifestations of sarcomere mutations in dilated cardiomyopathy (DCM) in comparison with sarcomere mutations associated with HCM [24]. Also, this study was cross-sectional, and compared normal genotype-negative subjects, overt-sarcomeric DCM, sarcomeric-subclinical DCM and sarcomeric-subclinical HCM. The authors showed that subtle systolic abnormalities were identified in subclinical DCM-mutation carriers, while impaired relaxation and preserved systolic function were the predominant early manifestations of sarcomeric HCM [24]. Similarly, a study in children carrying HCM-causing mutations found signs of diastolic dysfunction in about half of the carriers [25].

The main conclusion of all these studies is that carriers of HCM mutations show subtle morphologic or functional cardiac abnormalities even in the absence of LVH [26]. However, there are HCM carriers who will never develop the phenotype and it is still unclear whether the subclinical abnormalities found at baseline will ultimately result in the manifestation of clinically significant disease in individual genetic carriers. For the first time, we analyzed the data from the same HCM carriers at baseline and, after a long follow up, demonstrated that in the single subjects some abnormal echocardiographic parameters at baseline may predict the development of the phenotype.

Our findings highlight the concept that HCM is fundamentally a disease of sarcomere dysfunction, with poor sarcomere relaxation being a central pathogenic mechanism [10]. The resultant increase in cardiomyocyte tension can impair myocardial deformation velocities, observable during isovolumic relaxation. The abnormal baseline GLS, despite normal LV ejection fraction, in Gen+/Phen− individuals who later develop LVH, may be caused by underlying myocardial disarray and reduced perfusion [27]. These abnormalities are known to precede structural changes such as LVH and myocardial fibrosis, which are hallmarks of HCM [7].

This study is pioneering in comparing baseline echocardiographic parameters in HCM Gen+/Phen− subjects who develop LVH with those who do not. Identifying age- and load-independent predictors like GLS and SRIVR has significant clinical implications. These parameters are sensitive and easily obtainable, and allow the early detection of myocardial dysfunction in mutation carriers. This early detection is crucial, as it opens avenues for timely intervention, potentially slowing or preventing the progression to overt HCM.

Although there are currently no studies in humans demonstrating that drugs can slow the progression of subclinical HCM [28], basic science studies continue to add knowledge on the pathogenesis of the disease [10]. These studies suggest that new therapeutic strategies could potentially halt the common molecular phenotype of hyperdynamic contractility, poor relaxation, and increased energy consumption already present in Gen+/Phen− individuals. These fundamental pathophysiologic anomalies are manifested early at a subclinical level in Gen+/Phen− individuals, and drive the hypertrophic remodeling that produces overt HCM [29].

These findings highlight the importance of following sarcomere variant carriers longitudinally and the critical need to improve understanding of factors that drive disease penetrance and progression. Implementing preventive strategies in genetically predisposed individuals before the appearance of LVH could have beneficial effects on the disease trajectory [10,30]. For instance, tailored medical therapy aimed at improving myocardial relaxation and reducing cardiomyocyte tension might be beneficial. Additionally, lifestyle modifications and regular monitoring using these sensitive echocardiographic markers can be part of a comprehensive management plan for at-risk individuals.

It is important to note that our findings are specific to Gen+/Phen− subjects and do not suggest that the echocardiographic abnormalities we found could be used to identify sarcomeric mutations in patients who already present with LVH. The utility of these findings is primarily in the early detection and monitoring of Gen+/Phen− subjects, allowing for more personalized and proactive medical care. Furthermore, we have no data to suggest the predictive value of GLS and SRIVR in sudden-death stratification or the role of these echocardiographic parameters in the differential diagnosis of other forms of LVH. Further studies may be conducted in larger populations with LVH to establish the role of GLS and SRIVR as potential biomarkers.

Larger studies are required to integrate genetic, clinical, and echocardiographic data to generate a scoring system more reliable than any individual factor alone in predicting the phenotypic expression of the disease. Furthermore, longitudinal studies assessing the impact of early therapeutic interventions based on these predictive markers would provide robust evidence for their clinical application. Additionally, exploring the molecular mechanisms underlying the observed echocardiographic changes can enhance our understanding of HCM pathogenesis and identify new therapeutic targets.

## 4. Methods

### 4.1. Study Population

The Gen+/Phen− study cohort was selected from the database of the HCM Outpatient Clinic of the Federico II University of Naples, Italy, over an enrollment period of about 20 years (from 2002 to 2023). The clinical diagnosis of familial HCM was established as recommended by the current guidelines [31], by the presence of LVH defined in presence of an LV maximal wall thickness (MWT) ≥ 13 mm in subjects ≥ 18 years of age or an MWT z-score relative to body surface area > 2 in subjects < 18 years of age [31]. MWT was the biggest thickness measured from LV short-axis views at mitral-valve, papillary-muscle and apical levels. Subjects were eligible for inclusion in the study if they (1) were confirmed to carry the proband pathogenic HCM mutation; (2) had a normal standard echocardiogram at the first visit; and (3) did not show intra-, inter-ventricular or atrio–ventricular blocks. The final study population consisted of sixteen subjects (25% women, age 20 ± 11 years, range 8–37 years) enrolled during family screening, after the identification of the causal mutation in the proband. Baseline and follow-up cardiological visits and echocardiograms were available for all study subjects. LVH at follow-up was defined in presence of MWT of at least 13 mm. Informed consent was obtained from all participants in protocols approved by the Ethics Committee of the Federico II University of Naples, Italy (101_2015).

### 4.2. Echocardiography

The 2D and Doppler echocardiograms selected for the study were performed with the Philips IE33 system in the gene carriers on 2 occasions: at the time of the first evaluation, when there was no evidence of the phenotype, and at the last follow-up which was the date of the last echocardiogram for the subjects who did not develop the phenotype, or the date of the first follow-up echocardiogram showing evidence of LVH in those who developed the phenotype.

Images were stored digitally and analyzed offline by two expert sonographers, unaware of the clinical history, independently. Standard measures of cardiac dimensions were determined as the mean on 3 cardiac cycles. Echocardiography was performed as previously described [32,33]. Briefly, the end-diastolic thicknesses of the interventricular septum (IVS) and posterior wall (PW) were measured from 2-dimensional parasternal long-axis images at the level of the mitral valve leaflet tips and at the papillary-muscle level. In addition to measuring the IVS and PW, the uniformity of LV wall thickness was visually assessed in all LV segments by examining the images taken at multiple parasternal short-axis levels (basal, mid, and apical), and from the apical views, to identify regions with disproportionately increased thickness. If a relatively thickened segment was identified, the location was noted and the MWT was measured. The LV ejection fraction (EF) was calculated by the biplane modified Simpson’s rule. Mitral inflow was recorded at the tips of the mitral leaflets by Pulsed Wave (PW) Doppler and the velocities at the peaks of early, passive filling (E) and during atrial contraction (A) were measured. Tissue Doppler imaging (TDI) was performed as follows: Pulsed-Doppler was applied to the septal and lateral corners of the mitral annulus to allow for a spectral display of the velocities at these 2 points. Systolic (S’), early diastolic (E’), and late diastolic (A’) velocities were measured. The E’/A’ and E/E’ ratios were computed at both corners of the mitral annulus. Gains and filters were adjusted carefully to eliminate background noise and allow for a clear tissue signal. Longitudinal strain was obtained from apical images (4 chamber-, 2 chamber- and 3-chamber views). During image acquisition for longitudinal strain, frame rates were maximized by narrowing the sector to isolate individual walls (range 50 to 70 frames per second). Off-line image analysis was performed using the TomTec software Version 4.4 (TOMTEC Imaging Systems GmbH, Unterschleissheim, Germany) with speckle tracking methodology, which tracks the movement of natural acoustic speckles in the myocardium from 2D gray-scale images [34,35,36].The endocardium was manually traced, and myocardial motion was tracked by the automated software. Tracking quality was verified manually and with the software’s automated quality-grading scale. Segments were rejected if adequate quality could not be obtained despite manual or automatic corrections. LV global strain and strain rate (SR) in each view were calculated using the entire length of the LV myocardium. Isovolumic relaxation (IVR) period was selected by measuring the time from the aortic valve closure to the mitral valve opening on the recorded electrocardiogram. Peak global SR during the IVR period (SRIVR) was computed as the peak positive curvature identified during the IVR period on the SR curves [21]. The global SR values from the 3 apical views were also averaged and used for final analysis.

### 4.3. Genetics

A blood sample was collected in EDTA. Genomic DNA was extracted using the Maxwell 16 instrument (Promega, Madison, WI, USA). DNA quality was assessed by Nanodrop spectrophotometer (Thermo Fisher, Waltham, MA, USA) and Tape Station analyzer (Agilent, Santa Clara, CA, USA). In the probands, targeted Next-Generation Sequencing (NGS) was performed by using a custom-made panel including 60 genes known to be associated with genetic cardiomyopathies, including HCM, dilated and arrhythmogenic cardiomyopathies. Libraries were prepared by using the HaloPlex technology (Agilent, Santa Clara, CA, USA) and subsequently sequenced using the MiSeq (Illumina, San Diego, CA, USA) instrument (2 × 250 PE). The Alyssa software (Agilent, Santa Clara, CA, USA) was used to align the sequences to the reference genome and obtain a list of genomic variants. Variants were described based on the current Human Genome Variation Society mutation nomenclature guidelines using the accession number of the longest transcript, and the variants were classified according to the current American College of Medical Genetics (ACMG) recommendations [34,37].

Pathogenic and likely pathogenic variants were confirmed by Sanger sequencing. Once the causal mutation was identified in the proband, the presence of the mutation was tested in the first-degree family members by Sanger sequencing.

### 4.4. Statistics

Data were analyzed using SPSS (version 25.0; IBM-SPSS, Armonk, NY, USA). Continuous variables were described as mean ± standard deviation. Categorical variables were described as number (percentage). An unpaired T test was used to test differences at baseline between subjects who developed or did not develop LVH at follow-up. A paired T test was also run when echocardiographic changes between baseline and follow-up were tested in each of the two groups of subjects. The χ2 test was used to compare categorical variables, with the Monte Carlo simulation to obtain exact *p*-values. Correlation of age with diastolic indexes was performed by Pearson correlation. Intra- (GC) and interobserver (GC and CS) repeatability coefficient was determined as 1.96× standard deviation of the absolute value of the differences in 10 consecutive subjects [33]. A value of *p* < 0.05 was considered to be significant.

## 5. Conclusions

Our study highlights the importance of advanced echocardiographic techniques in sub-clinical assessment of HCM when the clinically significant/relevant disease burden/symptomatology are still absent. The early detection of myocardial dysfunction using parameters like SRIVR and GLS in Gen+/Phen− individuals who later develop LVH suggests the potential for these markers in guiding early preventative strategies. This proactive approach could significantly improve outcomes for individuals at risk of developing HCM, transforming the current clinical paradigm and paving the way for personalized medicine in cardiomyopathy care.

## Figures and Tables

**Figure 1 ijms-25-08128-f001:**
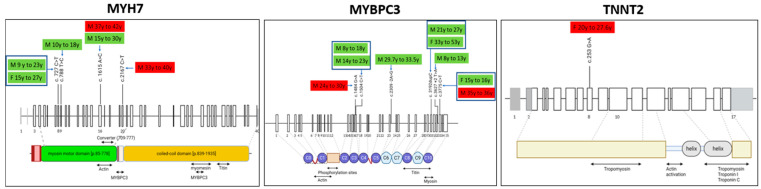
Schematic representation of the distribution of mutations in *MYH7*, *MYBPC3* and *TNNT2* genes in our population. The corresponding protein domains with their relative function and interactions are also shown. Green rectangle: phenotype-negative at follow-up; red rectangle: phenotype-positive at follow-up; F = female sex; M = male sex; y = years (age at baseline and at follow-up are shown).

**Figure 2 ijms-25-08128-f002:**
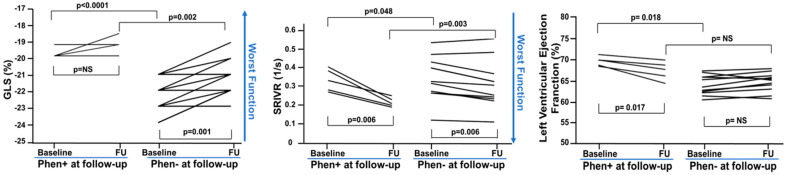
Global longitudinal strain (GLS), peak global strain rate during the isovolumic relaxation period (SRIVR) and ejection fraction (EF) variations during the follow up in Phen+ and Phen−. At baseline, GLS and EF were higher, and SRIVR lower in Phen+ as compared to Phen−. Over time, GLS and SRIVR worsened in both groups, while EF went down in Phen+, although it was still in the normal range.

**Figure 3 ijms-25-08128-f003:**
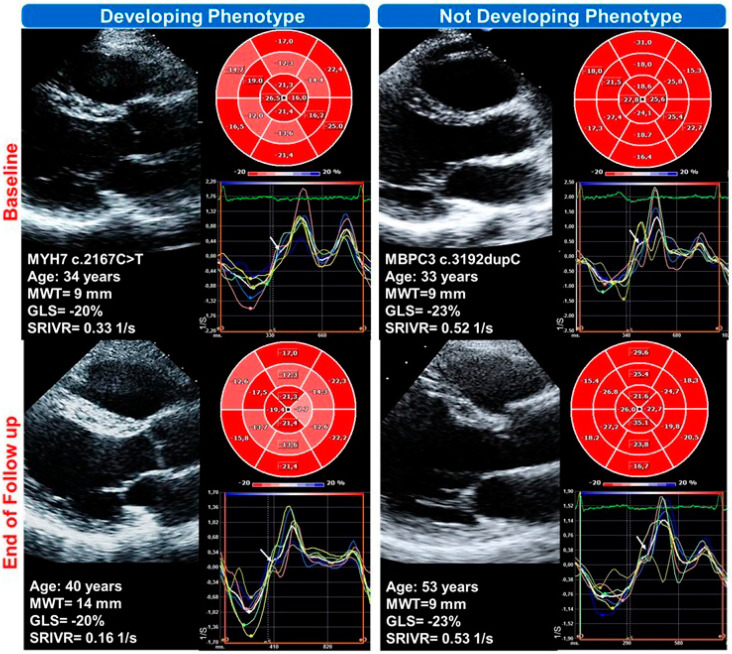
Examples of a Gen+/Phen− at baseline developing phenotype during follow-up (**left panels**) and a Gen+/Phen− at baseline not developing phenotype at follow-up (**right panels**). Higher global longitudinal strain (GLS) and lower peak global strain rate during the isovolumic relaxation period (SRIVR) at baseline suggest reduced systolic deformation and diastolic dysfunction, despite the absence of LVH, in the carriers who will develop the phenotype.

**Table 1 ijms-25-08128-t001:** Characteristics of the studied population.

Patient ID	Sex	Age at First Evaluation (Years)	Family N	Relationship with the Proband	Gene	Mutation	Age at Last Evaluation (Years)	Phenotype + at FU
AB	m	34	1	Son	MYH7	c.2167C > T (p.R723C)	40	yes
CA	m	9	2	Son	MYH7	c.788T > C (p.I263T)	18	no
CL	m	24	3	Son	MYBPC3	c.1484G > A (p.R495Q)	30	yes
DMC	f	20	4	Daughter	TNNT2	c.253G > A (p.V85M)	27	yes
DML	m	29	5	Son	MYBPC3	c.2309-2A > G (splicing)	33	no
EA	m	37	6	Brother	MYH7	c.1615A > C (p.M539V)	42	yes
EM	m	15	7	Son	MYH7	c.1615A > C (p.M539V)	29	no
FF	m	9	8	Son	MYH7	c.727C > T (p.R243C)	23	no
FS	f	15	8	Daughter	MYH7	c.727C > T (p.R243C)	26	no
MA	f	15	9	Daughter	MYBPC3	c.3775 C > T (p.Q1259X) stop	16	no
MR	m	35	9	Brother	MYBPC3	c.3775 C > T (p.Q1259X) stop	36	yes
NC	m	8	10	Son	MYBPC3	c.1504C > T (p.R502W)	18	no
NV	m	14	10	Son	MYBPC3	c.1504C > T (p.R502W)	23	no
UD	m	8	11	Son	MYBPC3	c.3627 + 2T > A (splicing)	13	no
VC	f	33	12	Sister	MYBPC3	c.3192dupC (p.K1065QfsX12)	54	no
VP	m	12	12	Son	MYBPC3	c.3192dupC (p.K1065QfsX12)	18	no

F = female sex; FU = follow-up; m = male sex; MYBPC3 = myosin-binding protein C3; MYH7 = myosin heavy chain 7; TNNT2 = troponin T2.

**Table 2 ijms-25-08128-t002:** Baseline differences between HCM gen carriers who developed and those who did not develop the HCM phenotype. Phen− = phenotype-negative; Phen+ = phenotype-positive.

Variable	Phen− at Follow Up (N = 11)	Phen+ at Follow-Up (N = 5)	*p*
Female sex (%)	27	20	0.755
Age at first evaluation (years)	15 ± 8	30 ± 8	0.005
Maximal wall thickness (mm)	8 ± 1.1	10 ± 0.5	0.025
Left ventricular ejection fraction (%)	64 ± 4	67 ± 2	0.096
E/E’ septal	7.2 ± 1.4	9.2 ± 1.5	0.023
E/E’ lateral	6.1 ± 1.6	6.4 ± 1.4	0.685
E’/A’ septal	1.7 ± 0.6	1.3 ± 0.8	0.246
E’/A’ lateral	2.0 ± 0.6	1.7 ± 0.7	0.324
Global longitudinal strain (%)	−22.3 ± 1.1	−19.8 ± 0.4	<0.0001
Strain rate during isovolumic relaxation (1/s)	0.40 ± 0.11	0.28 ± 0.5	0.048

**Table 3 ijms-25-08128-t003:** Follow-up differences between subjects who developed or did not develop the HCM phenotype. LV = left ventricular; Phen− = phenotype-negative; Phen+ = Phenotype-positive.

	Phen− at Follow Up (N = 11)	Phen+ at Follow-Up (N = 5)	*p*
Age at last evaluation (years)	25 ± 11	35 ± 6	0.076
Follow-up (years)	9.5 ± 5.5	5.2 ± 2.5	0.120
Maximal wall thickness (mm)	9 ± 0.8	13 ± 0.4	<0.0001
LV ejection fraction (%)	63 ± 4	62 ± 6	0.752
E’ septal (cm/s)	12 ± 3	8 ± 4	0.029
E’ lateral (cm/s)	17 ±5	13± 2	0.167
A’ septal (cm/s)	8± 2	9 ± 3	0.288
A’ lateral (cm/s)	7 ± 2	11 ± 3	0.020
S’ Septal (cm/s)	9 ± 1	8 ± 5	0.730
S’ lateral (cm/s)	10 ± 1	11 ± 2	0.242
E/E’ septal	8.8 ± 3.2	10.2 ± 0.9	0.350
E/E’ lateral	6.1 ± 0.9	6.6 ± 1.7	0.418
E’/A’ septal	1.8 ± 0.5	1.1 ± 0.4	0.025
E’/A’ lateral	2.4 ± 1.0	1.3 ± 0.6	0.035
GLS (%)	−21.2 ± 1.2	−19.0 ± 0.7	0.002
SRIVR (1/s)	0.37 ± 0.12	0.17 ± 0.0	0.003
GLS Changes > − 0.56% (#, %)	8, 73	3, 60	0.611
Changes SRIVR > 0.03 1/s (#, %)	5, 54.5	5, 100	0.069

GLS = global longitudinal strain; SRIVR = peak global SR during the IVR period.

## Data Availability

Data are contained within the article.

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
