# Peer review of "Echocardiographic Strain Abnormalities Precede Left Ventricular Hypertrophy Development in Hypertrophic Cardiomyopathy Mutation Carriers"

_ijms, 2024, doi:10.3390/ijms25158128_

Round 1

Reviewer 1 Report

Comments and Suggestions for Authors

This reviewer thanks the authors for their comprehensive original article evaluating the clinical relationship between genotype and variable expressivity in the setting of sarcomere protein point mutations for a small cohort size. A number of fundamental concerns limit this reviewer's ability to provide a complete evaluation of the article, the most pertinent of which are summarized below:

1. The abstract does not contain sufficient detail/descriptors for the reader to understand that this is a clinical study.

2. The authors repeatedly refer to their study in the abstract and body a "preclinical" (example: 301), however it seems that the study is clinical in nature, given it involves patients, as opposed to preclinical, which can refer to animal models of human pathophysiology. Though preclinical is a descriptor meant to refer to "prior to manifesting with clinically significant/relevant disease burden/symptomatology", this likely should be defined within the article to avoid confusion.

Furthermore, it may prove prudent to adopt a more-specific descriptor that has been utilized in the literature, namely "subclinical". Please see the below article:

Tardiff JC. It's never too early to look: subclinical disease in sarcomeric dilated cardiomyopathy. Circ Cardiovasc Genet. 2012 Oct 1;5(5):483-6. doi: 10.1161/CIRCGENETICS.112.964817. PMID: 23074334; PMCID: PMC4401148.

3. The authors are missing a character in the abbreviation of "genotype-positive/phenotype-negative", defined in line 52.

4. The "(#11)" and "(#5)" present in the first row of Tables 1 and 2 are unclear.

5. It is not clear after reading the article that the authors explicitly stated a formal definition of LVH. This is paramount given the stratification of patients at follow-up based on this variable.

6. The reviewer asks the authors to clearly state how their submission's findings differ from the below original articles and review article:

Ho CY, Carlsen C, Thune JJ, Havndrup O, Bundgaard H, Farrohi F, Rivero J, Cirino AL, Andersen PS, Christiansen M, Maron BJ, Orav EJ, Køber L. Echocardiographic strain imaging to assess early and late consequences of sarcomere mutations in hypertrophic cardiomyopathy. Circ Cardiovasc Genet. 2009 Aug;2(4):314-21. doi: 10.1161/CIRCGENETICS.109.862128. Epub 2009 Jun 19. PMID: 20031602; PMCID: PMC2773504.

Lakdawala NK, Thune JJ, Colan SD, Cirino AL, Farrohi F, Rivero J, McDonough B, Sparks E, Orav EJ, Seidman JG, Seidman CE, Ho CY. Subtle abnormalities in contractile function are an early manifestation of sarcomere mutations in dilated cardiomyopathy. Circ Cardiovasc Genet. 2012 Oct 1;5(5):503-10. doi: 10.1161/CIRCGENETICS.112.962761. Epub 2012 Sep 4. PMID: 22949430; PMCID: PMC3646896.

Ho CY, Day SM, Colan SD, Russell MW, Towbin JA, Sherrid MV, Canter CE, Jefferies JL, Murphy AM, Cirino AL, Abraham TP, Taylor M, Mestroni L, Bluemke DA, Jarolim P, Shi L, Sleeper LA, Seidman CE, Orav EJ; HCMNet Investigators. The Burden of Early Phenotypes and the Influence of Wall Thickness in Hypertrophic Cardiomyopathy Mutation Carriers: Findings From the HCMNet Study. JAMA Cardiol. 2017 Apr 1;2(4):419-428. doi: 10.1001/jamacardio.2016.5670. PMID: 28241245; PMCID: PMC5541992.

Poutanen T, Tikanoja T, Jääskeläinen P, Jokinen E, Silvast A, Laakso M, Kuusisto J. Diastolic dysfunction without left ventricular hypertrophy is an early finding in children with hypertrophic cardiomyopathy-causing mutations in the beta-myosin heavy chain, alpha-tropomyosin, and myosin-binding protein C genes. Am Heart J. 2006 Mar;151(3):725.e1-725.e9. doi: 10.1016/j.ahj.2005.12.005. PMID: 16504640.

Ho CY. Hypertrophic cardiomyopathy: preclinical and early phenotype. J Cardiovasc Transl Res. 2009 Dec;2(4):462-70. doi: 10.1007/s12265-009-9124-7. Epub 2009 Sep 26. Erratum in: J Cardiovasc Transl Res. 2013 Aug;6(4):662. PMID: 20560004.

The authors' claim to novelty seems to be over-stated or misinformed. Additional information or context is necessary.

Comments on the Quality of English Language

A few minor grammar errors are present, mostly relating to verb-subject conjugation as well as conjugation/prepositions. Careful review of the article following these revisions by an English language scholar is recommended.

Author Response

We appreciate your thoughtful comments and valuable suggestions to improve impact our paper. In this letter, we used black for your comments, black for our answers, and italic black to report sentences from the paper.

REVIEWER 1

This reviewer thanks the authors for their comprehensive original article evaluating the clinical relationship between genotype and variable expressivity in the setting of sarcomere protein point mutations for a small cohort size.

We thank the Reviewer for Her/His consideration about our manuscript.

A number of fundamental concerns limit this reviewer's ability to provide a complete evaluation of the article, the most pertinent of which are summarized below:

  1. The abstract does not contain sufficient detail/descriptors for the reader to understand that this is a clinical study.

We agree on the point raised by the Reviewer. Accordingly, we have changed the first sentences of the Abstract (Page 1, lines17-21) as follows:

Hypertrophic cardiomyopathy (HCM) is a genetic disease characterized by unexplained left ventricular hypertrophy (LVH), diastolic dysfunction, and increased sudden death risk. Early detection of the phenotypic expression of the disease in genetic carriers without LVH (Gen+/Phen-) is crucial for emerging therapies. This clinical study aims to identify echocardiographic predictors of phenotypic development in Gen+/Phen-.

  1. The authors repeatedly refer to their study in the abstract and body a "preclinical" (example: 301), however it seems that the study is clinical in nature, given it involves patients, as opposed to preclinical, which can refer to animal models of human pathophysiology. Though preclinical is a descriptor meant to refer to "prior to manifesting with clinically significant/relevant disease burden/symptomatology", this likely should be defined within the article to avoid confusion. Furthermore, it may prove prudent to adopt a more-specific descriptor that has been utilized in the literature, namely "subclinical". Please see the below article: Tardiff JC. It's never too early to look: subclinical disease in sarcomeric dilated cardiomyopathy. Circ Cardiovasc Genet. 2012 Oct 1;5(5):483-6. doi: 10.1161/CIRCGENETICS.112.964817. PMID: 23074334; PMCID: PMC4401148.

We agree with the Reviewer's point. Consequently, we have replaced the term "preclinical" with the suggested sentence "prior to manifesting with clinically significant/relevant disease burden/symptomatology", and made slight adjustments to fit the context within the text. In addition, we introduced the term used by Tardiff JC et al. in their article, i.e. “subclinical”, and, accordingly we added the reference You have suggested.

Below are the changes we made in the manuscript to address the Reviewer’s concerns:

  • Abstract, page 1, lines 29-31

This is the first HCM study investigating subjects before they manifest clinically significant or relevant disease burden or symptomatology, comparing at baseline HCM Gen+/Phen- subjects who will develop LVH to those who will not

  • Discussion, Page 7, line 234; Page 8, lines 235-236

These fundamental pathophysiologic anomalies are manifested early at a subclinical level in Gen+/Phen- individuals and drive hypertrophic remodeling that produces overt HCM

Conclusions, Page 10, lines 349-351

Our study highlights the importance of advanced echocardiographic techniques in sub-clinical assessment of HCM when the clinically significant/relevant disease burden/symptomatology are still absent

  1. The authors are missing a character in the abbreviation of "genotype-positive/phenotype-negative", defined in line 52.

We thank the reviewer for her/his profound attention to our paper. We have corrected the abbreviation.

On page 2, line 59 (Gen+/Phen-).

  1. The "(#11)" and "(#5)" present in the first row of Tables 1 and 2 are unclear.

We used the symbol  # to indicate “number of subjects” in each group. We understand that could be misleading. So, we changed it in the tables by indicating that 11 and 5 were the number of subjects in each group, as follows:

Phen- at follow up (N=11) and Phen+ at follow-up (N=5)

  1. It is not clear after reading the article that the authors explicitly stated a formal definition of LVH. This is paramount given the stratification of patients at follow-up based on this variable.

We are sorry if our definition of LVH in HCM was not clear enough. We added a sentence in the introduction and changed our methods as follow.

1.Introduction. Page 1, lines 41-43

Accordingly with the guidelines (REF), being the LV hypertrophy (LVH) typically asymmetrical in HCM, the presence of LVH is defined by the presence of a maximal wall thickness (MWT)≥13 mm in the familial forms.

Methods: 4.1. Study population.  Page 8, lines 263-277

The Gen+/Phen- study cohort was selected from the database of the HCM Outpatient Clinic of the Federico II University of Naples, Italy, over an enrollment period of about 20 years (from 2002 to 2023). The clinical diagnosis of familial HCM was established as recommended by the current guidelines, by the presence of LVH defined in presence of a LV maximal wall thickness (MWT) ≥13 mm in subjects ≥18 years of age or a MWT z-score relative to body surface area >2 in subjects<18 years of age. MWT was the biggest thickness measured form LV short axis views at mitral valve, papillary muscles and apical levels. Subjects were eligible for inclusion in the study if they (1) were confirmed to carry the pro-band pathogenic HCM mutation; (2) had a normal standard echocardiogram at the first visit; (3) did not show intra, inter ventricular or atrio-ventricular blocks. The final study population consisted of sixteen subjects (25% women, age 20±11 years, range 8-37 years) enrolled during family screening, after the identification of the causal mutation in the proband. Baseline and follow-up cardiological visits and echocardiograms were available in all study-subjects. LVH at follow-up was defined in presence of MWT of at least 13 mm.

  1. The reviewer asks the authors to clearly state how their submission's findings differ from the below original articles and review article:

Ho CY, Carlsen C, Thune JJ, Havndrup O, Bundgaard H, Farrohi F, Rivero J, Cirino AL, Andersen PS, Christiansen M, Maron BJ, Orav EJ, Køber L. Echocardiographic strain imaging to assess early and late consequences of sarcomere mutations in hypertrophic cardiomyopathy. Circ Cardiovasc Genet. 2009 Aug;2(4):314-21. doi: 10.1161/CIRCGENETICS.109.862128. Epub 2009 Jun 19. PMID: 20031602; PMCID: PMC2773504.

Lakdawala NK, Thune JJ, Colan SD, Cirino AL, Farrohi F, Rivero J, McDonough B, Sparks E, Orav EJ, Seidman JG, Seidman CE, Ho CY. Subtle abnormalities in contractile function are an early manifestation of sarcomere mutations in dilated cardiomyopathy. Circ Cardiovasc Genet. 2012 Oct 1;5(5):503-10. doi: 10.1161/CIRCGENETICS.112.962761. Epub 2012 Sep 4. PMID: 22949430; PMCID: PMC3646896.

Ho CY, Day SM, Colan SD, Russell MW, Towbin JA, Sherrid MV, Canter CE, Jefferies JL, Mrphy AM, Cirino AL, Abraham TP, Taylor M, Mestroni L, Bluemke DA, Jarolim P, Shi L, Sleeper LA, Seidman CE, Orav EJ; HCMNet Investigators. The Burden of Early Phenotypes and the Influence of Wall Thickness in Hypertrophic Cardiomyopathy Mutation Carriers: Findings From the HCMNet Study. JAMA Cardiol. 2017 Apr 1;2(4):419-428. doi: 10.1001/jamacardio.2016.5670.; PMID: 28241245 PMCID: PMC5541992.

Poutanen T, Tikanoja T, Jääskeläinen P, Jokinen E, Silvast A, Laakso M, Kuusisto J. Diastolic dysfunction without left ventricular hypertrophy is an early finding in children with hypertrophic cardiomyopathy-causing mutations in the beta-myosin heavy chain, alpha-tropomyosin, and myosin-binding protein C genes. Am Heart J. 2006 Mar;151(3):725.e1-725.e9. doi: 10.1016/j.ahj.2005.12.005. PMID: 16504640.

Ho CY. Hypertrophic cardiomyopathy: preclinical and early phenotype. J Cardiovasc Transl Res. 2009 Dec;2(4):462-70. doi: 10.1007/s12265-009-9124-7. Epub2009 Sep 26. Erratum in: J Cardiovasc Transl Res. 2013 Aug;6(4):662. PMID: 20560004.

The authors' claim to novelty seems to be over-stated or misinformed. Additional information or context is necessary.

We thank the Reviewer for Her/His suggestion to highlight how our findings differ from similar published papers in subclinical HCM. The main difference between our study and the previously published studies is that the latter are all cross sectional-observational, while our study, to the best of our knowledge, is the first to retrospectively look at the same subjects over a long period of time to identify the baseline subclinical abnormalities which predicted the development of overt HCM.

Hence, we added the following sentences to the discussion to describe the findings of the previous studies (mentioned by the Reviewer) and ultimately explain the significance and novelty of our findings. Page 7, lines 187-213

Previous studies aimed to understand echocardiographic abnormalities in Gen+/Phen- subjects. Ho CY et al. performed a cross-sectional study, comparing TDI and strain echocardiographic data among normal subjects, Gen+/Phen- individuals, and overt HCM patients. The differences the authors identified among the three groups were of great interest yet did not clarify the characteristics that could help clinicians to predict phenotype development in Gen+/Phen- subjects. Another similar cross-sectional study focused on MWT and TDI has compared normal subjects, Gen+/Phen- individuals, and overt HCM patients. The authors found that greater LVMWT was associated with more prominent cardiac abnormalities in a continuous, although not always linear manner, and that a single value of LVMWT could not dichotomize the presence or absence of the disease. Another study characterized the early manifestations of sarcomere mutations in dilated cardiomyopathy (DCM) in comparison with sarcomere mutations associated with HCM. Also, this study was cross-sectional and compared normal genotype negative subjects, overt-sarcomeric DCM, sarcomeric-subclinical DCM and sarcomeric-subclinical HCM. The authors showed that subtle systolic abnormalities were identified in subclinical DCM mutation carriers, while impaired relaxation and preserved systolic function were the pre-dominant early manifestations of sarcomeric HCM24. Similarly, a study in children carrying HCM-causing mutations, found signs of diastolic dysfunction in about half of the carriers.

The main conclusion of all these studies is that carriers of HCM mutations show subtle morphologic or functional cardiac abnormalities even in the absence of LVH. However, there are HCM carriers who will never develop the phenotype and it is still unclear whether the subclinical abnormalities found at baseline will ultimately result in the manifestation of clinically significant disease in individual genetic carriers. For the first time, we analyzed the data from the same HCM carriers at baseline and after a long follow up and demonstrated that in the single subjects some abnormal echocardiographic parameters at baseline may predict the development of the phenotype.

A few minor grammar errors are present, mostly relating to verb-subject conjugation as well as conjugation/prepositions. Careful review of the article following these revisions by an English language scholar is recommended.

We thank the reviewer for her/his attention in our manuscript. We hopefully corrected overall the grammar errors.

Reviewer 2 Report

Comments and Suggestions for Authors

This is a novel retrospective study of 16 Genotype+/Phenotype- (Gen+/Phen-) carriers of hypertrophic cardiomyopathy (HCM) with 2 mutations responsible for 75% of the HCM cases known to have an identified genetic cause.  The Authors assessed a battery of echocardiographic parameters to determine which ones were predictive of progression to left ventricular hypertrophy (Phen+) at ~8 year follow-up.  The authors found that two echocardiographic strain abnormalities that were predictive of which Gen+/Phen- patients went on to have the phenotype of LVH independent of patient age and loading conditions: global longitudinal strain (GLS) and strain rate isovolumic relaxation (SRIVR).

This study is important because early detection of the HCM phenotype is potentially life-saving as these patients are at high risk of sudden cardiac death.  Such patients are thought to benefit from ICD placement and may be candidates for clinical trials of experimental therapies to slow progression of the disease.

Questions for Authors:

1)    Are the changes in GLS and SRIVR seen in the patients who went on to develop LVH the same changes one would expect in early-onset LVH from causes other than HCM?  I.e., are the echocardiographic findings seen in this study that were predictive of HCM-induced LVH also predictive of LVH in general or are these findings potentially specific to HCM-induced LVH?  The question is important because it changes the generalizability of the findings.  I.e., a holy grail of HCM screening would be an easily obtainable biomarker that can identify which patients with LVH (a humongous population) have HCM (a small subset of LVH patients at risk of sudden death that require very close monitoring and targeted therapy). 

2)    Given that multiple echocardiographic parameters were associated with progression from Gen+/Phen- -> Gen+/Phen+, did the Authors consider creating a scoring system involving multiple parameters that could predict this progression better than any individual factor alone?

Author Response

We appreciate your thoughtful comments and valuable suggestions to improve impact our paper. In this letter, we used black for your comments, black for our answers, and italic black to report sentences from the paper.

Reviewer 2

This is a novel retrospective study of 16 Genotype+/Phenotype- (Gen+/Phen-) carriers of hypertrophic cardiomyopathy (HCM) with 2 mutations responsible for 75% of the HCM cases known to have an identified genetic cause.  The Authors assessed a battery of echocardiographic parameters to determine which ones were predictive of progression to left ventricular hypertrophy (Phen+) at ~8 year follow-up.  The authors found that two echocardiographic strain abnormalities that were predictive of which Gen+/Phen- patients went on to have the phenotype of LVH independent of patient age and loading conditions: global longitudinal strain (GLS) and strain rate isovolumic relaxation (SRIVR).

This study is important because early detection of the HCM phenotype is potentially life-saving as these patients are at high risk of sudden cardiac death.  Such patients are thought to benefit from ICD placement and may be candidates for clinical trials of experimental therapies to slow progression of the disease.

We thank the Reviewer for Her/His positive evaluation of our study.

 Questions for Authors:

  • Are the changes in GLS and SRIVR seen in the patients who went on to develop LVH the same changes one would expect in early-onset LVH from causes other than HCM? e., are the echocardiographic findings seen in this study that were predictive of HCM-induced LVH also predictive of LVH in general or are these findings potentially specific to HCM-induced LVH?  The question is important because it changes the generalizability of the findings.  I.e., a holy grail of HCM screening would be an easily obtainable biomarker that can identify which patients with LVH (a humongous population) have HCM (a small subset of LVH patients at risk of sudden death that require very close monitoring and targeted therapy).

We thank the Reviewer for this consideration. We are not sure that our study focusing on HCM mutation carriers could be generalized to the population of patients with LVH. However, we think this is an important point, so we added a paragraph in the discussion section. Page 8, Lines 245-253.

It is important to note that our findings are specific to Gen+/Phen- subjects and do not suggest that the echocardiographic abnormalities we found could be used to identify sarcomeric mutations in patients who already present with LVH. The utility of these findings is primarily in the early detection and monitoring of Gen+/Phen- subjects, allowing for more personalized and proactive medical care. Furthermore, we have no data to suggest the predictive value of GLS and SRIVR in sudden death stratification or the role of these echocardiographic parameters in the differential diagnosis with other forms of LVH. Further studies may be conducted in larger populations with LVH to establish the role of GLS and SRIVR as potential biomarkers.

  • Given that multiple echocardiographic parameters were associated with progression from Gen+/Phen- -> Gen+/Phen+, did the Authors consider creating a scoring system involving multiple parameters that could predict this progression better than any individual factor alone?

The Reviewer suggestion is extremely interesting; however, our cohort is too small to generate a scoring system which will require a larger population and a validation cohort. Thus, we added the following sentence in the discussion section. Page 8, lines 254-256.

Larger studies are required to integrate genetic, clinical, and echocardiographic data to generate a scoring system more reliable than any individual factor alone in predicting the phenotypic expression of the disease.

Round 2

Reviewer 1 Report

Comments and Suggestions for Authors

This reviewer thanks the authors for their thorough revisions and responses. The remaining suggestions are listed below:

1. Ideally the abstract only contains the exact p values for each major statistic without the including exact means or standard deviations/errors, however this reviewer yields if another reviewer has instructed you otherwise

2. The article keywords should be different from the title to improve searchability and potential readership

3. The reviewer requests that the authors state precisely which analyses underwent unpaired versus paired student's t tests. Furthermore, these statistical details should be included in each figure legend for clarity.

4. Please define all variables in each table's respective legend for clarity.

5. The "Funding" section title (line 368) contains an extra "o"

Author Response

We appreciate your thoughtful comments and valuable suggestions to improve impact our paper. In this letter, we used black for your comments, black for our answers, and italic black to report sentences from the paper.

This reviewer thanks the authors for their thorough revisions and responses. The remaining suggestions are listed below:

  1. Ideally the abstract only contains the exact p values for each major statistic without the including exact means or standard deviations/errors, however this reviewer yields if another reviewer has instructed you otherwise.

Thank you. The other reviewer did not ask for elimination of values in the abstract, thus within the abstract we left the values of means and standard deviation

  1. The article keywords should be different from the title to improve searchability and potential readership

We would like to express our gratitude to the reviewer for their efforts in helping us improve our paper. We added other keywords, Page 1, lines 34-35

 Hypertrophic cardiomyopathy; screening; genetics; subclinical detection; strain echocardiography; global longitudinal strain; diastolic strain rate; left ventricular hypertrophy

  1. The reviewer requests that the authors state precisely which analyses underwent unpaired versus paired student's t tests. Furthermore, these statistical details should be included in each figure legend for clarity.

Again, we thank the reviewer for this point. Which allows us to better clarify how we handled the t test. We added sentences within the statistics section. Page 10, lines 343-345

Unpaired T test was used to test differences at baseline between subjects who developed or not LVH at follow-up. Paired T test was otherwise run when echocardiographic changes between baseline and follow-up were tested in each of the two groups of subjects.

  1. Please define all variables in each table's respective legend for clarity.

Thank you again. Table 1 has already a legend; In table 2 there were no acronyms, whereas Table 2 the legend was missed. Thus, we added a legend to Table 3, Page 5, line 150

GLS= global longitudinal strain; LV= Left ventricular; SRIVR= Peak global SR during the IVR period

  1. The "Funding" section title (line 368) contains an extra "o"

Thank you. We are very sorry. We corrected accordingly.

Round 3

Reviewer 1 Report

Comments and Suggestions for Authors

This reviewer yields.